# Indications of the *SERPINE 1* variant rs1799768's role in anti-VEGF therapy resistance in neovascular age-related macular degeneration

**Muhammer Özgür ÇEVİK** [1]*, **Zühal Mert Altuntaş**[2], **Sadık Görkem Çevik**[3]

**1** Department of Medical Genetics, Faculty of Medicine, Adiyaman University, Adiyaman, Türkiye,
**2** Department of Medical Genetics, Faculty of Medicine, Mersin University, Mersin, Türkiye, **3** Department of Ophthalmology, Beyoglu Eye Research Hospital Intravitreal Surgery Unit, University of Health Sciences, Istanbul, Türkiye

* ocevik@adiyaman.edu.tr

## Abstract

Age-related macular degeneration (AMD) is a retinal disease prevalent in the elderly population, with two main subtypes: dry (non-exudative) and neovascular (wet or exudative). Neovascular AMD (nAMD) has a more debilitating prognosis than dry AMD, making it the third leading cause of blindness. Intravitreal injections of anti-vascular endothelial growth factor (IV anti-VEGF) are the most effective and widely accepted treatment for nAMD. However, a significant number of nAMD patients exhibit suboptimal responses to IV anti-VEGF therapy, with the underlying mechanisms not yet fully understood. We hypothesized that genetic polymorphisms associated with blood hypercoagulation may also contribute to suboptimal responses to IV anti-VEGF therapy.

This study recruited 20 nAMD patients, who were divided into two groups based on their treatment responses after four years: 10 patients with suboptimal responses to IV anti-VEGF therapy and 10 patients with optimal responses. After obtaining institutional ethics board approval, we retrospectively evaluated relevant clinical records of twenty patients diagnosed with nAMD. Patient clinical data were accessed between 20th March 2021 -1st April 2021 for research purposes only. We genotyped peripheral blood DNA from each patient for hypercoagulation-related polymorphisms, including Factor V Leiden (rs6025), prothrombin c.20210G>A (rs1799963), MTHFR A1298C (rs1801131), MTHFR C677T (rs1801133), and SERPINE 1 (PAI-1-675 4G/5G) (rs1799768), and statistically compared the frequencies.

Heterozygous and homozygous mutations in the *SERPINE1* gene specifically PAI-1 promoter region PAI-1-675 4G/5G (rs1799768) were identified as risk factors for resistance to IV anti-VEGF therapy in nAMD patients ($\chi^2$ test, p = 0.006). No other polymorphisms of the above-mentioned genes were statistically significant (p > 0.05).

**Data availability statement:** All relevant data are within the manuscript and its Supporting Information files.

**Funding:** The author(s) received no specific funding for this work.

The failure of IV anti-VEGF therapy in nAMD patients may be influenced by various factors, one of which may be the inherited PAI-1-675 4G/5G (rs1799768) polymorphisms which normally known to contribute hypercoagulation. Further research involving a larger cohort is necessary to uncover the interplay between hereditary factors and other elements contributing to the inefficacy of IV anti-VEGF therapy in nAMD.

## Introduction

Age-related macular degeneration (AMD) is a chronic, debilitating condition prevalent among the elderly, primarily caused by damage to the macula—a 5.5 mm diameter pigmented region near the center of the retina crucial for spatial resolution and overall vision [1,2]. AMD is the leading cause of blindness in high-income countries [3] and is typically categorized into three stages: early, intermediate, and late [4]. Late-stage is when AMD causes the most significant vision impairment. Late-stage AMD is further divided into two subtypes: dry AMD (also known as atrophic, non-exudative, or non-neovascular) and neovascular AMD (nAMD, also called wet or exudative) [5]. Approximately 10–15% of late-stage AMD cases progress to nAMD, which accounts for 90% of blindness cases in the elderly due to irreversible vision loss [6]. nAMD can be treated with photodynamic therapy with verteporfin (PDTV) [7] and intravitreal injections of anti-vascular endothelial growth factor (anti-VEGF) therapies, although anti-VEGF monotherapies have become the golden standard of care for the last two decades. However, a significant proportion of treated patients exhibit suboptimal responses or develop tachyphylaxis over time [8]. This resistance to treatment has substantial socio-economic implications, and current theoretical frameworks fail to fully explain its underlying mechanisms [9].

To address this critical knowledge gap, the present study aims to investigate single nucleotide polymorphisms (SNPs) associated with coagulation and vascular processes. The focus includes Factor V Leiden (FVL 1691 G>A, rs6025), Factor II G20210A (prothrombin G20210A, rs1799963), MTHFR A1298C (rs1801131), MTHFR C677T (rs1801133), and SER-PINE1 (PAI-1-675 4G/5G, rs1799768). By analyzing the prevalence of these SNPs in nAMD patients with optimal and suboptimal responses to anti-VEGF treatment, our study seeks to elucidate whether these SNPs are associated with treatment efficacy.

## Material and methods

This study was conducted in accordance with the tenets of the Declaration of Helsinki after obtaining approval from the Bursa Yüksek İhtisas Hastanesi Ethics Committee Bursa,Türkiye (Ethics Board protocol number: 2021/03-03) and is a retrospective analysis of medical records, and the need for informed consent was waived by the ethics committee. Following institutional approval, we reviewed the clinical records of 20 patients diagnosed with neovascular age-related macular degeneration (nAMD). Data collection spanned from March 20, 2021, to April 1, 2021, and was used exclusively for research purposes.

As age-related macular degeneration predominantly affects the elderly, no minors were included in this study. Patients with comorbidities such as diabetes, hypertension, or cancer were excluded to minimize confounding factors. The study focused on 20 eyes from 20 patients with nAMD, all treated with intravitreal injections of anti-vascular endothelial growth factor (anti-VEGF) agents, specifically ranibizumab and aflibercept. Based on their response to therapy, patients were divided into two equal groups: Optimal responders and suboptimal responders, with 10 patients in each group (5 males and 5 females per group).

Optimal responders were defined as those showing a reduction in exudation, intraretinal or subretinal fluid, or maximum pigment epithelial detachment (PED) height following anti-VEGF therapy. Suboptimal responders were characterized by persistent or increased fluid and exudation for at least six months despite receiving intravitreal injections every four weeks.

All patients followed a treat-and-extend (TREX) regimen after completing their loading doses. If necessary, the type of VEGF inhibitor was switched after six doses. Choroidal neo-vascularization (CNV) parameters, including central macular thickness (CMT), maximum fluid height, and maximum PED height, were evaluated using Optical Coherence Tomography (OCT, Optovue RTVue, Fremont, California) and Fundus Angiography (FA, Topcon-900, Tokyo, Japan).

To investigate genetic factors, all patients were genotyped for SNPs namely Factor V Leiden (G1691A), Factor II (G20210A), MTHFR (A1298C and C677T), and SERPINE1 (PAI-1-675 4G/5G) with a Roche Lighcycler 480 RT-PCR machine. Genotyping for Factor V Leiden and Factor II mutations was conducted using the Factor V Leiden Kit and Factor II (Prothrombin) G20210A Kit (Roche Molecular Systems, Branchburg, NJ, USA). Positive controls for these mutations were validated using the LightMix Kit (TIB Molbiol GmbH, Berlin, Germany). MTHFR A1298C, MTHFR C677T, and PAI-1-675 4G/5G mutations were analyzed using the Light Cycler Fast Start DNA Master HybProbe (Roche Diagnostics GmbH, Mannheim, Germany). The Light Cycler Fast Start DNA Master HybProbe is a real-time PCR-based system that utilizes fluorescently labeled probes to detect and quantify specific DNA sequences. This approach allows for rapid and sensitive detection of genetic variants, making it a suitable technique for the analysis of the PAI-1-675 4G/5G polymorphism [10–12]. The above-mentioned commercial single-nucleotide polymorphism (SNP) genotyping kits are validated against known positive and controls to ensure the reliability of results.

## Statistical analysis

Data were expressed as mean ± standard deviation for parametric variables, and as percentages or frequencies for non-parametric variables. Statistical significance was defined as $p < 0.05$. Crosstabs were created to compare the presence of polymorphisms between the resistant and responsive groups, with data represented as number (n), percent (%), and chi-square ($\chi^2$). The effect of gene presence in Group 1 (optimal) and Group 2 (suboptimal) was analyzed using logistic regression, with results expressed as odds ratio (OR) and 95% confidence interval (CI). Statistical analyses and calculations were performed using IBM SPSS Statistics 21.0 (IBM Corp., Released 2012, Version 21.0, Armonk, NY: IBM Corp.) and MS-Excel 2007 by a professional licensed statistician. Statistical significance was set at $p < 0.05$.

## Results

The mean age of patients in the optimal response group was 72.2 ± 7.3 years (range: 60 to 86 years), while the mean age in the suboptimal response group was 73.8 ± 8.3 years (range: 61 to 88 years) (Table 1 and table in S1 Table). There was no statistically significant difference in age or gender distribution between the two groups (p > 0.05) (Table 1 and S1 Table). The mean follow-up period was 18 ± 3.5 months (range: 12 to 26 months) for the optimal response group and 23 ± 3.3 months (range: 17 to 29 months) for the suboptimal response group. The mean number of intravitreal injections was 6.6 ± 2.8 (range: 3 to 10) in the optimal response group and 13.2 ± 3.2 (range: 8 to 17) in the suboptimal response group, with a statistically significant difference between the two groups (p=0.01 <0.05) (Table 1). No statistically significant difference was observed in intimal thickness between consecutive injections within the resistant group (p > 0.05). A sample of consecutive injections for a single patient is presented in Fig 1.

**Table 1. Demographics. Demographics and treatment features of participants.**

|  | Patients (n) | Gender |  | Age (years) | Follow-up (months) | Average number of consecutive injections |
|---|---|---|---|---|---|---|
|  |  | Female | Male |  |  |  |
| **Optimal** | 10 | 5 | 5 | 72.2 ± 7.3 | 18 ± 3.5 | 6.6 ± 2.8 |
| **Suboptimal** | 10 | 5 | 5 | 73.8 ± 10.3 | 23 ± 3.3 | *13.2 ± 3.2 |

*p=0.01 <0.05

We genotyped the DNA of all nAMD patients for Factor V Leiden (FVL 1691G > A, rs6025), Factor II G20210A (prothrombin G20210A, rs1799963), MTHFR A1298C (rs1801131), MTHFR C677T (rs1801133), and *SERPINE 1* (PAI-1-675 4G/5G) (rs1799768). The results are summarized in Table 2 and S2 Table. We found a statistically significant difference between the optimal and suboptimal response groups (p=0.006<0.05) regarding only the *SERPINE 1* (PAI-1-675 4G/5G) rs1799768 polymorphism, with the suboptimal response group showing a significantly higher frequency of this mutation. No other polymorphisms studied were correlated to anti-VEGF resistance. No *Factor II* G20210A mutations were observed in either group (0%). The *Factor V* Leiden G1691A mutation was present only in the heterozygous form in two patients (20%) in the resistant group and was absent (0%) in the responsive group.

Although not statistically significant, the *MTHFR* C677T allele was more commonly observed in the optimal responsive group (six patients: five heterozygous-50% and one homozygous-10%) than the responsive group (two patients: 20%, heterozygous form). Similarly, *MTHFR* A1298C mutations were relatively more frequent in the optimal responsive group (six patients: five heterozygous [50%], one homozygous-10%) than in the responsive group (three patients: 30%). In addition, the homozygous *SERPINE 1* (PAI-1-675 4G/4G) polymorphism was more prevalent in resistant patients (40%) compared to responsive patients (0%). Conversely, the wild-type *SERPINE 1 PAI-1-675* 5G/5G was observed four times more common in the responsive group (80%) than in the resistant group (20%). The heterozygous *SERPINE 1* (PAI-1-675 4G/5G) was present in 20% of the responsive group versus 10% of the resistant group. The presence of PAI-1-675 4G/5G mutations in either the heterozygous or homozygous form was significantly higher in the resistant group (80%, n=8) compared to the responsive group (20%, n=2) (p=0.001).

Logistic regression analysis was conducted to assess the association between polymorphisms and suboptimal response to anti-VEGF therapy (Table 3). The analysis revealed no statistically significant associations between gene variants and suboptimal response although the *SERPINE 1* polymorphism yielded the closest trend toward association without reaching statistical significance with an odds ratio (OR) of **0.125** (95% CI: 0.013–1.245) and a p-value of **0.076**. No significant associations were found for *FV Leiden (G1691A)* (OR = 0.001, 95% CI: 0.001–0.001, p = 0.999), *MTHFR (C677T)* (OR = 0.200, 95% CI: 0.026–1.526, p = 0.121), or *MTHFR (A1298C)* (OR = 0.343, 95% CI: 0.052–2.261, p = 0.266) and results were farer from significance compared to *SERPINE 1*.

## Discussion

The incomplete response to anti-VEGF therapy in nAMD is a significant clinical challenge, necessitating further exploration of underlying mechanisms and potential biomarkers. Hence, we have chosen SNP genotyping method in order to contribute to this knowledge gap as untangling complex diseases have greatly benefited from the SNP genotyping studies [13]. In this study, we identified a statistically significant association between having

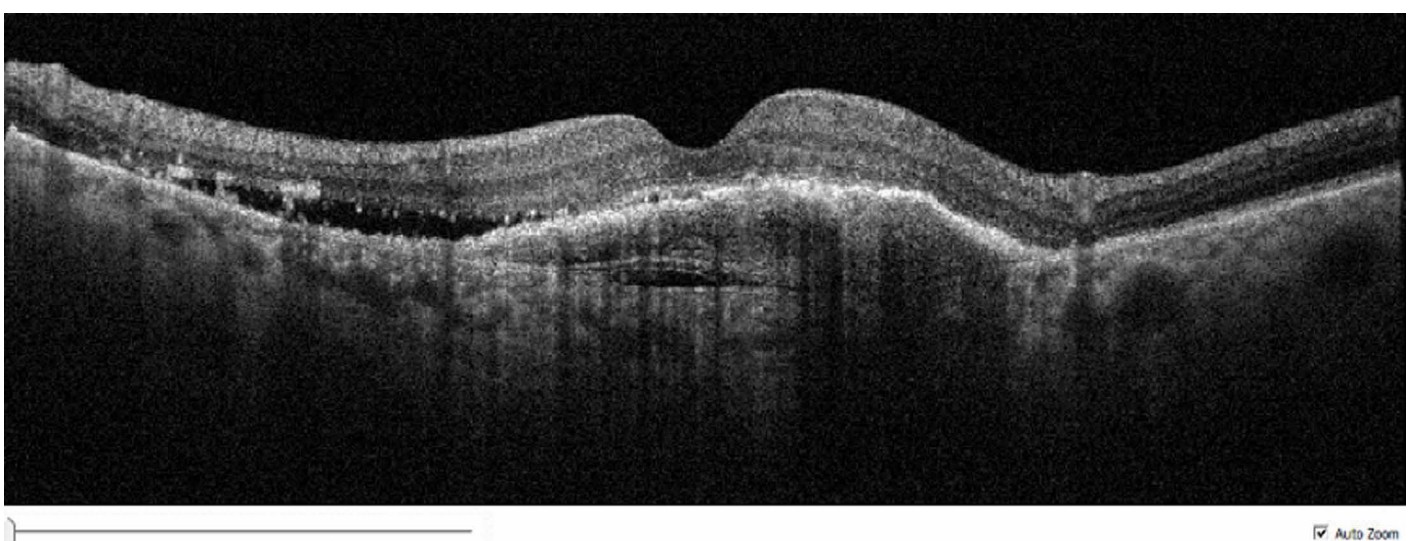

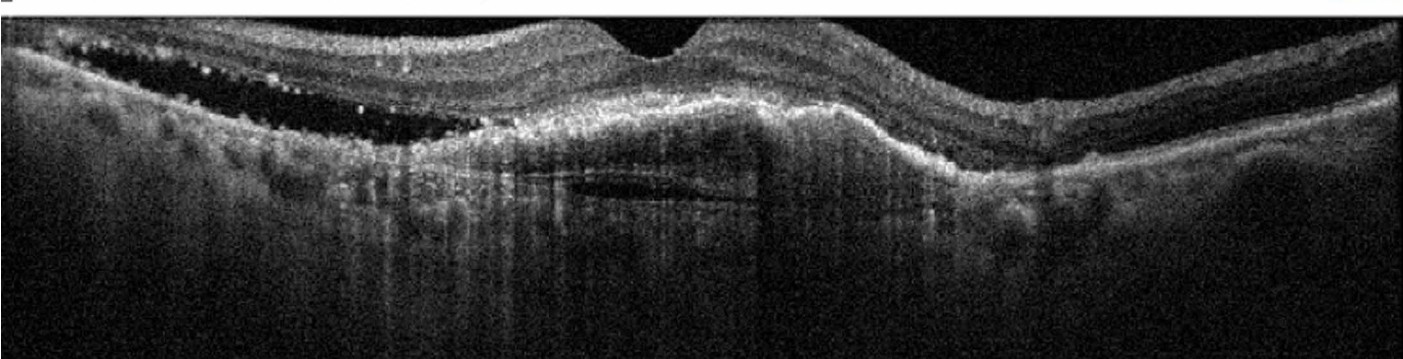

**Fig 1. Representative photo of OCT measurement.** A sample of consequent intimal thickness OCT photos of IV-anti-VEGF applied suboptimal responsive patient.

a PAI-1 4G polymorphism and suboptimal response to anti-VEGF therapy, suggesting a potential role for this SNP in modulating treatment responses of nAMD patients. This aligns with previous literature reporting that nAMD is associated with increased levels of plasma PAI-1 [14] although we have not measured any PAI-1 protein levels. Other SNPs analyzed in our study, including Factor V Leiden (G1691A), MTHFR (C677T), and MTHFR (A1298C), showed no statistically significant associations. This contrasts with previous reports that aforementioned SNPs were correlated with suboptimal responses to PDTV in nAMD patients. This phenomenon has been associated with the presence of genetic variants such as Factor V Leiden, Factor II G20210A, MTHFR A1298C, MTHFR C677T [15], and SERPINE1 PAI-1-675 4G/5G [16]. Insights into the mechanisms underlying PDTV resistance prompted us to investigate the same SNPs for their potential role in resistance to anti-VEGF therapies in nAMD. At the end, our findings suggest that the mechanisms driving anti-VEGF resistance in nAMD may differ from those involved in PDTV resistance since logistic regression analysis did not reveal statistically significant associations for any of the studied polymorphisms, including rs1799768, although a trend was observed for the PAI-1 polymorphism (p = 0.076). This finding indicates a potential role for the PAI-1 polymorphism [17,18] which warrants further investigation in studies with larger sample sizes.

**Table 2. SNP Genotyping and Anti-VEGF Response Comparison Results. Genotyping and chi-square (χ2) comparison results of the nAMD patients treated with anti-VEGF drugs.**

|  | OPTIMAL RESP. | SUBOPTIMAL RESP. | χ2 | P |
|---|---|---|---|---|
| **FV LEIDEN (G1691A)** |  |  |  |  |
| Normal | 8 (80.0) | 10 (100) | - | 0.474** |
| Heterozygous | 2 (20.0) | 0 |  |  |
| Homozygous | 0 | 0 |  |  |
| **FII (G20210A)** | 10 (100.0) | 10 (100.0) |  |  |
| Normal | 0 | 0 | - | - |
| Heterozygous | 0 | 0 |  |  |
| Homozygous | 0 | 0 |  |  |
| **MTHFR (C677T)** |  |  |  |  |
| Normal | 4 (40.0) | 8 (80.0) | 4.074 | 0.130* |
| Heterozygous | 5 (50.0) | 2 (20.0) |  |  |
| Homozygous | 1 (10.0) | 0 (0.0) |  |  |
| **MTHFR (A1298C)** |  |  |  |  |
| Normal | 4 (40.0) | 7 (70.0) | 2.72 | 0.257* |
| Heterozygous | 5 (50.0) | 3 (30.0) |  |  |
| Homozygous | 1 (10.0) | 0 |  |  |
| **SERPINE 1 (PAI-1-675 4G/5G)** |  |  |  |  |
| Normal | 2 (20.0) | 8 (80.0) | 10.08 | 0.006* |
| Heterozygous | 4 (40.0) | 2 (20.0) |  |  |
| Homozygous | 4 (40.0) | 0 |  |  |

*Likelyhood ratio.

**Fisher exact test

**Table 3. Regression Analysis Results. Logistic regression analysis of detected variants in both responsive and sub-responsive groups.**

|  | OR | 95% CONFIDENCE INTERVAL | P |
|---|---|---|---|
| **FV LEIDEN (G1691A)** | 0.001 | 0.001-0.001 | 0.999 |
| **FII (G20210A)** | NO VARIANT EXISTS |  |  |
| **MTHFR (C677T)** | 0.2 | 0.026-1.526 | 0.121 |
| **MTHFR (A1298C)** | 0.043 | 0.052-2.261 | 0.266 |
| **SERPINE 1 (PAI-1-675 4G/5G)** | 0.125 | 0.013-1.245 | 0.076 |

**OR: Odd's ratio**

PAI-1 belongs to the plasminogen activator inhibitor (PAI) family of proteins. There are at least three well-defined types of PAIs in the human body: PAI-1, PAI-2 and PAI-3, which are expressed by separate genes and serve distinct functions[14]. PAI-3 is a plasma protein C inhibitor, encoded by SERPINA5 gene and expressed in various organs [19]. PAI-2, encoded by the SERPINB2 gene, is primarily expressed in the placenta during fetal development and is rarely detected postnatally [20]. PAI-1, also called as the endothelial cell PAI, is encoded by the SERPINE1 gene (the PAI-1 gene), which is located on chromosome 7 (7q21.3-q22) [21]. The SERPINE1 gene has several important polymorphisms in its promoter region. Three of them —namely rs1799768, rs1799889, and rs1799762—have been well studied and are regarded as identical, despite being distinct genomic entries [13]. The rs1799768 is characterized by an insertion or deletion of a single guanine nucleotide at position -675 in the promoter region of

the gene [22]. This insertion results in two alleles: the 4G allele, which contains four consecutive guanine nucleotides, and the 5G allele, which contains five consecutive guanine nucleotides at this position [22]. There is no evidence to suggest that these SNPs affect the structure of the PAI-1 protein since mutation is located in the promoter region of the SERPINE1 gene. In addition, PAI-1 expression in circulation is relatively low and tightly regulated under normal physiological conditions [23]. On the other hand, mutations in the non-coding regions of genes have long been known to have the potential to affect gene expression [24]. Likewise, PAI-1 4G allele leads to enhanced transcription of PAI-1 protein, resulting in elevated PAI-1 levels in circulation [11,25]. Elevated levels of PAI-1 in circulation associated to various morbidities such as thrombotic disorders [26], myocardial infarction, metabolic syndrome [27], vascular complications [28], angiogenesis [29] and fibrinogenesis [30]. In the circulatory system, PAI-1 protein is produced by various cells, including endothelial cells, hepatocytes, adipocytes, platelets and plasma cells [31]. PAI-1 mainly inhibits tissue-type plasminogen activator (tPA) and urokinase (uPA), both of which activate plasminogen facilitate fibrinolysis and regulate blood clot stability [21]. PAI-1 is expressed in ocular tissues as well, including the retina [32], cornea [33], and aqueous humor [34], suggesting both systemic and ocular production of PAI-1. Moreover, in corneal epithelial cells, PAI-1 has been shown to act as an adhesion substrate and promote cell migration, suggesting a potential role in corneal wound healing [35]. This widespread distribution implies that PAI-1 may have local and systemic functions throughout different parts of the body and the overall homeostasis of the eye.

Monitoring the PAI-1 concentrations across different tissues and comparing these levels in various body compartments are essential for understanding the pathology of nAMD and the resistance to anti-VEGF treatments. To date, very few studies have directly compared PAI-1 levels in systemic circulation with those in the eye globe [36]. In one of those studies, patients with diabetic retinopathy had significantly higher serum PAI-1 levels than those without retinopathy. However, the levels of PAI-1 in tears did not show any significant difference between groups, suggesting that local biomarkers like tear PAI-1 levels may not be a good biomarker as serum PAI-1 levels [14]. Another study, which focuses on proliferative diabetic retinopathy, indicate that PAI-1 is significantly expressed in the vitreous and neovascular tissues of eyes affected by proliferative diabetic retinopathy (PDR), where its expression is regulated by hypoxia-inducible factor (HIF)-2α. This suggests a localized increase in PAI-1 levels in the eye compared to systemic circulation, contributing to retinal neovascularization [32]. Increased PAI-1 expression has also been observed in retinal microvessels of patients with nonproliferative diabetic retinopathy, implicating PAI-1 in the pathogenesis of diabetic eye complications [32]. Furthermore, in a model of oxygen-induced retinopathy, PAI-1 expression was upregulated during active angiogenesis, where at the same time PAI-1 null mice exhibited a significant reduction (53%) in retinal angiogenesis compared to wild-type mice, emphasizing PAI-1's importance in retinal angiogenesis process [32]. In addition, studies using transgenic mice overexpressing human PAI-1 have shown that these mice had increased PAI-1 levels in retinal micro-vessels, thickened basement membranes and alterations in endothelial cell-to-pericyte ratios [37].

In the context of suboptimal response to anti-VEGF in nAMD, increased levels of plasma PAI-1 was found to be associated with nAMD pathology [38]. In another study, authors compared aqueous humor samples of nAMD patients who were normal, responsive, and suboptimal responsive to anti-VEGF therapy [39]. The study detected an abundance of the fibrinogen alpha chain in the aqueous humor of suboptimal responders, suggesting that the fibrinolytic pathway may contribute to anti-VEGF treatment resistance [39]. Furthermore, continuous administration of an anti-VEGF-A antibody was shown to upregulate PAI-1 expression in endothelial cells, highlighting a potential link between elevated PAI-1 levels and anti-VEGF

resistance [40]. However, one should keep in mind that blood retinal barrier (BRB) is an important phenomenon in regulating ocular levels of proteins. BRB maintains retinal homeostasis by selectively regulating the movement of molecules between the systemic circulation and the retina with a selective permeability. This selective permeability can also adjust the levels and activity of PAI-1 in the ocular environment compared to systemic circulation [41]. On the other hand, abnormal PAI-1 levels due to any reason can lead to BRB breakdown as well, leading to retinal vascular leakage and edema [42]. Therefore, PAI-1 plays critical roles in ocular tissues and may influence responses to anti-VEGF therapy in nAMD through mechanisms that remain to be elucidated. Hence, genetic studies such as SNP genotyping or measuring circulating PAI-1 levels, may provide a more reliable biomarker strategy compared to local PAI-1 levels although this phenomenon needs further research.

Our study has several limitations. First, it is important to acknowledge that these findings establish an association, not causality, between the PAI-1 4G allele and resistance to anti-VEGF therapy in nAMD. Second, functional studies and signaling data are necessary to confirm whether the rs1799768 polymorphism leads to significant protein-level alterations that contribute to anti-VEGF resistance. Third, while we employed validated commercial kits with positive and negative controls, the generalizability of our findings to broader populations should be interpreted with caution. Since the retrospective study design, small cohort size, and exclusion of comorbidities further limit the applicability and generalization of our results. Fourth, we did not directly measure PAI-1 protein levels in biological fluids from the nAMD patients. Instead, our assumption of increased PAI-1 levels is based on our SNP genotyping results and existing medical literature on the PAI-1 4G polymorphism. Direct measurements of PAI-1 protein levels in larger and more diverse cohorts are necessary to validate this hypothesis.

While not directly reflecting our study's findings, we propose here a potential mechanism for the interactions between the potentially increased PAI-1 and VEGF pathways by using the related KEGG pathway database [43]. We hypothesize that PAI-1 4G SNP increases PAI-1 levels and this may contribute to the production of reactive oxygen species, akin to the effects observed in hyperglycemic conditions in the abovementioned studies. This heightened oxidative stress may result in the formation of advanced glycation end-products (AGEs), which activate pathways promoting VEGF overexpression. Ultimately, this process could drive pathogenic angiogenesis, further increasing VEGF production and complicating resistance to anti-VEGF therapy (Fig 2).

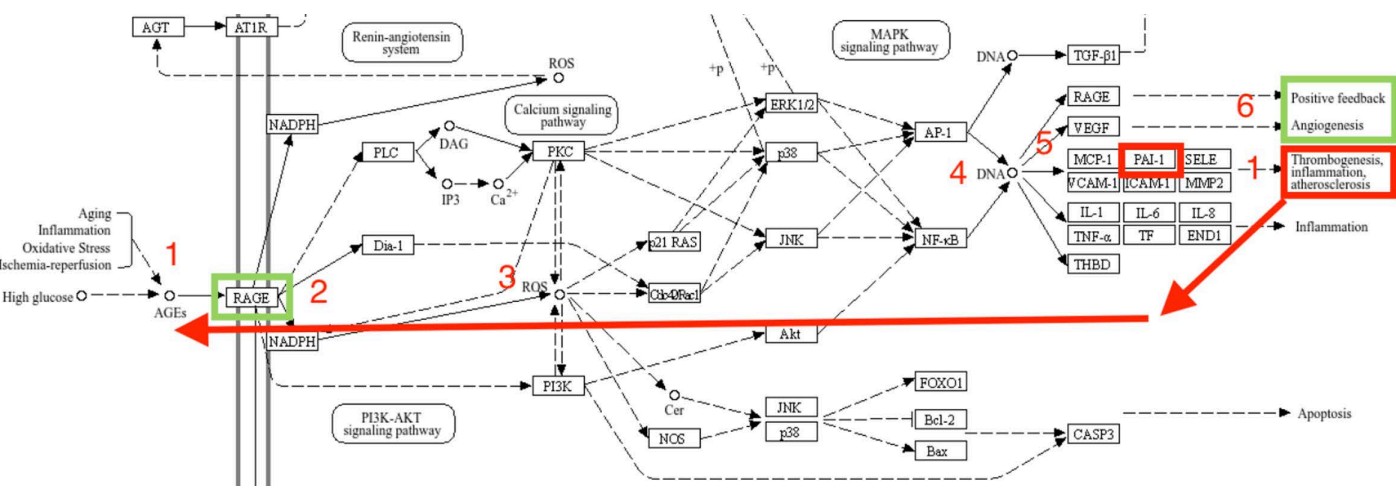

**Fig 2. Proposed mechanism for the role of PAI-1 in VEGF expression.**

As a conclusion, this research highlights the potential of PAI-1 4G SNP in anti-VEGF treatment of nAMD, contributing to future individualized patient care. The link between PAI-1 and anti-VEGF therapy resistance indicates that PAI-1 could be a promising therapeutic target in nAMD. Genetic profiling, could identify patients who are unlikely to benefit from standard anti-VEGF therapies, paving the way for targeted interventions. However, PAI-1's roles in ocular and circulatory system are multifaceted, comorbidities such as diabetes and endothelial disease can alter healthy PAI-1 levels in addition to bearing PAI-1 4G allele hereditarily. Future research with larger, more diverse cohorts and functional studies is critical for validating these findings, clarifying PAI-1 4G's role. Additionally, testing drugs that modulate PAI-1 levels for efficacy and safety in nAMD treatment is a necessary step forward. Combining anti-VEGF agents with PAI-1 modulators or other adjuvant therapies that target suboptimal response mechanisms may improve outcomes for nAMD patients.

## Supporting information

**S1 Table. Patients' ages, follow-up periods and injection times.** F indicates female patients, M indicates male patients.
(DOCX)

**S2 Table. Genotyping results.** 'Optimal' refers to responsive patients, 'suboptimal' refers to suboptimal responsive patients.
(DOCX)

## Acknowledgment and/or disclaimers

We would like to thank to state licensed biostatistician Emre Yaşar (Istanbul, Turkey) due to his diligent work for statistical analysis.

## Author contributions

**Conceptualization:** MUHAMMER OZGUR CEVIK, Zühal Mert Altuntaş, Sadik Gorkem Cevik.

**Data curation:** MUHAMMER OZGUR CEVIK, Zühal Mert Altuntaş.

**Formal analysis:** MUHAMMER OZGUR CEVIK.

**Investigation:** MUHAMMER OZGUR CEVIK, Sadik Gorkem Cevik.

**Methodology:** MUHAMMER OZGUR CEVIK, Zühal Mert Altuntaş.

**Project administration:** MUHAMMER OZGUR CEVIK, Zühal Mert Altuntaş.

**Resources:** MUHAMMER OZGUR CEVIK, Sadik Gorkem Cevik.

**Supervision:** MUHAMMER OZGUR CEVIK.

**Validation:** MUHAMMER OZGUR CEVIK.

**Visualization:** MUHAMMER OZGUR CEVIK.

**Writing – original draft:** MUHAMMER OZGUR CEVIK.

**Writing – review & editing:** MUHAMMER OZGUR CEVIK, Zühal Mert Altuntaş, Sadik Gorkem Cevik.

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
