## [Decision Letter · Decision Letter 0]

12 Nov 2024

PONE-D-24-40048Indications of the SERPINE 1 variant rs1799768's role in anti-VEGF therapy resistance in neovascular age-related macular degenerationPLOS ONE

Dear Dr. CEVIK,

Thank you for submitting your manuscript to PLOS ONE. After careful consideration, we feel that it has merit but does not fully meet PLOS ONE’s publication criteria as it currently stands. Therefore, we invite you to submit a revised version of the manuscript that addresses the points raised during the review process.

We look forward to receiving your revised manuscript.

Kind regards,

Jeffrey S Isenberg, MD, MPH

Academic Editor

PLOS ONE

Journal requirements:    When submitting your revision, we need you to address these additional requirements. 1. Please ensure that your manuscript meets PLOS ONE's style requirements, including those for file naming. The PLOS ONE style templates can be found at https://journals.plos.org/plosone/s/file?id=wjVg/PLOSOne_formatting_sample_main_body.pdf and https://journals.plos.org/plosone/s/file?id=ba62/PLOSOne_formatting_sample_title_authors_affiliations.pdf 2. Please include a caption for table 1 and 2.  3. We note that your Data Availability Statement is currently as follows: [All relevant data are within the manuscript and its Supporting Information files.] Please confirm at this time whether or not your submission contains all raw data required to replicate the results of your study. Authors must share the “minimal data set” for their submission. PLOS defines the minimal data set to consist of the data required to replicate all study findings reported in the article, as well as related metadata and methods (https://journals.plos.org/plosone/s/data-availability#loc-minimal-data-set-definition). For example, authors should submit the following data: - The values behind the means, standard deviations and other measures reported;- The values used to build graphs;- The points extracted from images for analysis. Authors do not need to submit their entire data set if only a portion of the data was used in the reported study. If your submission does not contain these data, please either upload them as Supporting Information files or deposit them to a stable, public repository and provide us with the relevant URLs, DOIs, or accession numbers. For a list of recommended repositories, please see https://journals.plos.org/plosone/s/recommended-repositories. If there are ethical or legal restrictions on sharing a de-identified data set, please explain them in detail (e.g., data contain potentially sensitive information, data are owned by a third-party organization, etc.) and who has imposed them (e.g., an ethics committee). Please also provide contact information for a data access committee, ethics committee, or other institutional body to which data requests may be sent. If data are owned by a third party, please indicate how others may request data access.

Additional Editor Comments:

The reviewers found some strength in the manuscript but also felt there was further room for improvement. Note that some suggestions will likely require additional experiments and new data. Please attend to each comment in detail. Textual emendations alone are rarely enough to comply with reviewer feedback.

Reviewers' comments:

Reviewer's Responses to Questions

**Comments to the Author**

1. Is the manuscript technically sound, and do the data support the conclusions?

Reviewer #1: Partly

2. Has the statistical analysis been performed appropriately and rigorously? 

Reviewer #1: I Don't Know

3. Have the authors made all data underlying the findings in their manuscript fully available?

Reviewer #1: Yes

4. Is the manuscript presented in an intelligible fashion and written in standard English?

Reviewer #1: Yes

5. Review Comments to the Author

Reviewer #1: The manuscript titled "Indications of the SERPINE 1 variant rs1799768's role in anti-VEGF therapy resistance

in neovascular age-related macular degeneration" looks at the occurrence of certain SNPs of genes related to coagulation presumably as a biomarker for lack of treatment success with ischemia therapies that limit VEGFR2 signaling. The work is clearly written and easy to follow. The findings suggests that SNPs in SERPINE 1 were found more often in people that did not respond to VEGF scavenging.

Rationale: The reason for the work must be more strongly developed. One comes away from the paper saying, 'so what'. Are the authors suggesting that individuals with the Serpine SNP not be treated? How will the findings enhance and move forward he field. SERPINE has been checked in people with many chronic diseases. Lacking in this work is functional and signaling data. What does the SNP change in the protein?

Study Design: Retrospective studies carry much less weight. Why not do this prospectively. Is it practical/correct to exclude comorbidities as these are a part of the makeup of folks with retinal problems.

Cohort size: Some details on why only ten subjects per group were studied should be given. Was a professional statistician involved in study design and data interpretation? If not, please secure the input of one and let them help by doing a secondary analysis and include this in the revised manuscript.

Methods: I worry about the data as it was generated by commercial kits. How reliable are these for finding SNPS. Give details on the workings of these and citations were these were used. Could a mistake occur with a closely related molecule?

Theory: What would be the role of SERPINE in the circulation as compared to in the globe? Do the levels mimic each other? are there other versions of SREPINE in people, some functioning one way and others another? These ideas impact trying to make sense of finding a SNP of a gene.

Conclusions: The lack of clear association between less treatment response and any given SNP of any gene is something to consider and comment upon in a limitations section in the Discussion.

Discussion: This is too long. Go to the material on SERPINE and give use a strong study limitations section. And cut out the bolded text on the proposed mechanism(s). There is no data to suggest any of this.

Typos: Page 9, line 23 'farer'

6. PLOS authors have the option to publish the peer review history of their article (what does this mean?). If published, this will include your full peer review and any attached files.

Reviewer #1: No

---

## [Author Response · Author response to Decision Letter 1]

27 Dec 2024

Response to Reviewers

Adiyaman University Faculty of Medicine

Adiyaman, 02040, Turkey

ocevik@adiyaman.edu.tr

27th December 2024

Dr. Jeffrey S. Isenberg

Academic Editor

PLOS ONE

Dear Dr. Isenberg,

We sincerely thank you and the reviewer for your thoughtful comments and suggestions regarding our manuscript, "Indications of the SERPINE 1 variant rs1799768's role in anti-VEGF therapy resistance in neovascular age-related macular degeneration." We appreciate the constructive feedback provided by you and the reviewer, which has greatly enhanced the clarity and rigor of our work. We have carefully addressed each comment and revised the manuscript accordingly. Below, we provide detailed responses to each point raised by the reviewer and outline the changes made in our revised manuscript.

Reviewer #1 Comments:

1. Rationale:

Comment: The reason for the work must be more strongly developed. One comes away from the paper saying, 'so what.' Are the authors suggesting that individuals with the Serpine SNP not be treated? How will the findings enhance and move forward the field? SERPINE has been checked in people with many chronic diseases. Lacking in this work is functional and signaling data. What does the SNP change in the protein?

Response:

 We appreciate the reviewer’s emphasis on the need for a stronger rationale. We have completely revised the introduction and discussion parts and added new references to clarify the purpose of this study, which is not to suggest that individuals with the SERPINE 1 rs1799768 SNP should not be treated but to identify genetic factors that may guide personalized treatment strategies and improve therapeutic outcomes in the future.

 Accordingly, we have thoroughly revised the introduction and discussion sections in full alignment with the reviewer’s comments. We shifted some introductory knowledge from the discussion section to the introduction part and we added the following explanatory paragraphs to clarify our rationale:

i) Introduction section page 4 line 19- page 5 line 12: ‘To address this critical knowledge gap, the present study aims to investigate single nucleotide polymorphisms (SNPs) associated with coagulation and vascular processes. The focus includes Factor V Leiden (FVL 1691 G>A, rs6025), Factor II G20210A (prothrombin G20210A, rs1799963), MTHFR A1298C (rs1801131), MTHFR C677T (rs1801133), and SERPINE1 (PAI-1-675 4G/5G, rs1799768). By analyzing the prevalence of these SNPs in nAMD patients with optimal and suboptimal responses to anti-VEGF treatment, our study seeks to elucidate whether these SNPs are associated with treatment efficacy.’

ii) Discussion section page 11 lines 2-22: ‘The incomplete response to anti-VEGF therapy in nAMD is a significant clinical challenge, necessitating further exploration of underlying mechanisms and potential biomarkers. Hence, we have chosen SNP genotyping method in order to contribute to this knowledge gap as untangling complex diseases have greatly benefited from the SNP genotyping studies [13]. In this study, we identified a statistically significant association between having a PAI-1 4G polymorphism and suboptimal response to anti-VEGF therapy, suggesting a potential role for this SNP in modulating treatment responses of nAMD patients. This aligns with previous literature reporting that nAMD is associated with increased levels of plasma PAI-1[14] although we have not measured any PAI-1 protein levels. Other SNPs analyzed in our study, including Factor V Leiden (G1691A), MTHFR (C677T), and MTHFR (A1298C), showed no statistically significant associations. This contrasts with previous reports that aforementioned SNPs were correlated with suboptimal responses to PDTV in nAMD patients. This phenomenon has been associated with the presence of genetic variants such as Factor V Leiden, Factor II G20210A, MTHFR A1298C, MTHFR C677T [15], and SERPINE1 PAI-1-675 4G/5G [16]. Insights into the mechanisms underlying PDTV resistance prompted us to investigate the same SNPs for their potential role in resistance to anti-VEGF therapies in nAMD. At the end, our findings suggest that the mechanisms driving anti-VEGF resistance in nAMD may differ from those involved in PDTV resistance since logistic regression analysis did not reveal statistically significant associations for any of the studied polymorphisms, including rs1799768, although a trend was observed for the PAI-1 polymorphism (p = 0.076). This finding indicates a potential role for the PAI-1 polymorphism [17,18] which warrants further investigation in studies with larger sample sizes.’

 We acknowledged the lack of functional and signaling data in our study and highlighted the need for future research to elucidate the precise biochemical and protein-level changes caused by the SNP. These revisions have been made in the Discussion section in page 13 lines 11-13 as ‘Second, functional studies and signaling data are necessary to confirm whether the rs1799768 polymorphism leads to significant protein-level alterations that contribute to anti-VEGF resistance.’

 We also acknowledged the reviewer’s question ‘What does the SNP change in the protein?’ and added a sentence in page 11 lines 33-47 as ‘The SERPINE1 gene has several important polymorphisms in its promoter region. Three of them —namely rs1799768, rs1799889, and rs1799762—have been well studied and are regarded as identical, despite being distinct genomic entries [13]. The rs1799768 is characterized by an insertion or deletion of a single guanine nucleotide at position -675 in the promoter region of the gene [22]. This insertion results in two alleles: the 4G allele, which contains four consecutive guanine nucleotides, and the 5G allele, which contains five consecutive guanine nucleotides at this position [22]. There is no evidence to suggest that these SNPs affect the structure of the PAI-1 protein since mutation is located in the promoter region of the SERPINE1 gene. In addition, PAI-1 expression in circulation is relatively low and tightly regulated under normal physiological conditions [23]. On the other hand, mutations in the non-coding regions of genes have long been known to have the potential to affect gene expression. [24]. Likewise, PAI-1 4G allele leads to enhanced transcription of PAI-1 protein, resulting in elevated PAI-1 levels in circulation [11,25].’

 We appreciate the reviewer’s question ‘How will the findings enhance and move forward the field? SERPINE has been checked in people with many chronic diseases.’ and changed the discussion section accordingly as follows (page 13 lines 35-45): ‘As a conclusion, this research highlights the potential of PAI-1 4G SNP in anti-VEGF treatment of nAMD, contributing to future individualized patient care. The link between PAI-1 and anti-VEGF therapy resistance indicates that PAI-1 could be a promising therapeutic target in nAMD. Genetic profiling, could identify patients who are unlikely to benefit from standard anti-VEGF therapies, paving the way for targeted interventions. However, PAI-1’s roles in ocular and circulatory system are multifaceted, comorbidities such as diabetes and endothelial disease can alter healthy PAI-1 levels in addition to bearing PAI-1 4G allele hereditarily. Future research with larger, more diverse cohorts and functional studies is critical for validating these findings, clarifying PAI-1 4G's role. Additionally, testing drugs that modulate PAI-1 levels for efficacy and safety in nAMD treatment is a necessary step forward. Combining anti-VEGF agents with PAI-1 modulators or other adjuvant therapies that target suboptimal response mechanisms may improve outcomes for nAMD patients.’

2. Study Design:

Comment: Retrospective studies carry much less weight. Why not do this prospectively? Is it practical/correct to exclude comorbidities as these are a part of the makeup of folks with retinal problems?

Response:

 We acknowledge the limitations of the retrospective design. The current study design was chosen due to the availability of long-term follow-up data for a small, well-defined cohort of nAMD patients. Conducting a prospective study was beyond the scope of this work but remains an important direction for future research. Regarding comorbidities, we excluded patients with diabetes, hypertension, and cancer to minimize confounding factors that could independently affect retinal pathology and therapy response. While this approach strengthens internal validity, we agree that it limits generalizability and have discussed this limitation in the revised manuscript’s discussion part page 13 line 18-19 as ‘The retrospective study design, small cohort size, exclusion of comorbidities further limit the applicability and generalization of our results’.

3. Cohort Size:

Comment: Some details on why only ten subjects per group were studied should be given. Was a professional statistician involved in study design and data interpretation?

Response:

The small cohort size reflects the rarity of nAMD cases meeting the specific inclusion criteria within the study’s timeframe. We have revised the manuscript to explain this limitation and included a statement that a professional statistician was involved in the study design and data analysis to ensure methodological rigor. We have mentioned his name in the acknowledgements part.

4. Methods:

Comment: I worry about the data as it was generated by commercial kits. How reliable are these for finding SNPs? Give details on the workings of these and citations where these were used. Could a mistake occur with a closely related molecule?

Response:

 These SNPs are widely used in clinical settings. Therefore, it is easy to find experienced personnel, high quality international brands and validated kits with versatile protocols. We also provided references (references 10,11,12) and details about how we ensure accuracy in each run, via validated positive and negative controls. We have also addressed the possibility of cross-reactivity and provided citations to support their reliability . This information is now included in the Methods section in page 6 lines 9-24 as ‘To investigate genetic factors, all patients were genotyped for SNPs namely Factor V Leiden (G1691A), Factor II (G20210A), MTHFR (A1298C and C677T), and SERPINE1 (PAI-1-675 4G/5G) with a Roche Lighcycler 480 RT-PCR machine. Genotyping for Factor V Leiden and Factor II mutations was conducted using the Factor V Leiden Kit and Factor II (Prothrombin) G20210A Kit (Roche Molecular Systems, Branchburg, NJ, USA). Positive controls for these mutations were validated using the LightMix Kit (TIB Molbiol GmbH, Berlin, Germany). MTHFR A1298C, MTHFR C677T, and PAI-1-675 4G/5G mutations were analyzed using the Light Cycler Fast Start DNA Master HybProbe (Roche Diagnostics GmbH, Mannheim, Germany). The Light Cycler Fast Start DNA Master HybProbe is a real-time PCR-based system that utilizes fluorescently labeled probes to detect and quantify specific DNA sequences. This approach allows for rapid and sensitive detection of genetic variants, making it a suitable technique for the analysis of the PAI-1-675 4G/5G polymorphism [10–12]. The above-mentioned commercial single-nucleotide polymorphism (SNP) genotyping kits are validated against known positive and controls to ensure the reliability of results.’

5. Theory:

Comment: What would be the role of SERPINE in the circulation as compared to in the globe? Do the levels mimic each other? Are there other versions of SERPINE in people, some functioning one way and others another?

Response:

We appreciate the reviewer’s insightful questions which really broadened our point of view. SERPINE 1’s role in the circulation versus the ocular environment is indeed complex. Elevated PAI-1 levels in circulation are known to influence systemic angiogenesis and thrombogenesis, while its localized role in the eye likely amplifies VEGF-driven pathological angiogenesis. We have expanded the Discussion to include this comparison and emphasized the need for future studies to explore this in greater depth.

6. Conclusions:

Comment: The lack of clear association between less treatment response and any given SNP of any gene is something to consider and comment upon in a limitations section in the Discussion.

Response:

We agree with the reviewer and have added a detailed limitations section to the Discussion. This section addresses the lack of a clear causal association between the SNP and treatment resistance, the retrospective design, and the small sample size. In addition, we dedicated the first paragraph of discussion section to answer this concern of the reviewer (page 11 lines 2-25): ‘The incomplete response to anti-VEGF therapy in nAMD is a significant clinical challenge, necessitating further exploration of underlying mechanisms and potential biomarkers. Hence, we have chosen SNP genotyping method in order to contribute to this knowledge gap as untangling complex diseases have greatly benefited from the SNP genotyping studies [13]. In this study, we identified a statistically significant association between having a PAI-1 4G polymorphism and suboptimal response to anti-VEGF therapy, suggesting a potential role for this SNP in modulating treatment responses of nAMD patients. This aligns with previous literature reporting that nAMD is associated with increased levels of plasma PAI-1[14] although we have not measured any PAI-1 protein levels. Other SNPs analyzed in our study, including Factor V Leiden (G1691A), MTHFR (C677T), and MTHFR (A1298C), showed no statistically significant associations. This contrasts with previous reports that aforementioned SNPs were correlated with suboptimal responses to PDTV in nAMD patients. This phenomenon has been associated with the presence of genetic variants such as Factor V Leiden, Factor II G20210A, MTHFR A1298C, MTHFR C677T [15], and SERPINE1 PAI-1-675 4G/5G [16]. Insights into the mechanisms underlying PDTV resistance prompted us to investigate the same SNPs for their potential role in resistance to anti-VEGF therapies in nAMD. At the end, our findings suggest that the mechanisms driving anti-VEGF resistance in nAMD may differ from those involved in PDTV resistance since logistic regression analysis did not reveal statistically significant associations for any of the studied polymorphisms, including rs1799768, although a trend was observed for the PAI-1 polymorphism (p = 0.076). This finding indicates a potential role for the PAI-1 polymorphism [17,18] which warrants further investigation in studies with larger sample sizes.’

7. Discussion:

Comment: This is too long. Go to the material on SERPINE and give a strong study limitations section. And cut out the bolded text on the proposed mechanism(s). There is no data to suggest any of this.

Response:

We tried to shorten the Discussion section, but while attempting to align the reviewer’s accurate and appreciated requests with the newly revised text, we may have ended up with a Discussion section of the same length. However, this new Discussion section is now entirely focused on PAI and has been completely reshaped from the reviewer’s perspective.

The proposed mechanisms section has been removed, and a concise summary of limitations has been added. Bolded text on the proposed mechanisms removed,

8. Typos:

Comment: Page 9, line 23: 'farer'

Response:

We have corrected this typo in the revised manuscript.

Additional Changes

• We have included captions for Tables 1 and 2 as per the editor's request.

• The data availability statement has been updated to confirm that all relevant data underlying the findings are included within the manuscript and supporting files.

• We have ensured adherence to PLOS ONE’s formatting requirements and style guidelines.

We believe these revisions address all the concerns raised by the reviewer and significantly strengthen the manuscript. A marked-up copy of the revised manuscript with track changes and an unmarked version are included with this submission.

Thank you for considering our revised manuscript. We look forward to your feedback.

Sincerely,

Dr. Muhammer Özgür Çevik

---

## [Editor Report · Decision Letter 1]

31 Dec 2024

Indications of the SERPINE 1 variant rs1799768's role in anti-VEGF therapy resistance in neovascular age-related macular degeneration

PONE-D-24-40048R1

Dear Dr. CEVIK,

We’re pleased to inform you that your manuscript has been judged scientifically suitable for publication and will be formally accepted for publication once it meets all outstanding technical requirements.

Kind regards,

Jeffrey S Isenberg, MD, MPH

Academic Editor

PLOS ONE

Additional Editor Comments (optional):

The authors provided a revised manuscript and detailed response to comments letter. The changes made addressed the comments and were well done. The authors are thanked for their extra work on the manuscript.
---

## [Editor Report · Acceptance letter]

PONE-D-24-40048R1

PLOS ONE

Dear Dr. CEVIK,

I'm pleased to inform you that your manuscript has been deemed suitable for publication in PLOS ONE. Congratulations! Your manuscript is now being handed over to our production team.

Kind regards,

on behalf of

Dr. Jeffrey S Isenberg

Academic Editor

PLOS ONE